# Text-to-graph Generation with Conditional Diffusion Models Guided by Graph-aligned LLMs

## Abstract

Text-to-graph generation, aiming for controlled graph generation based on natural language instructions, holds significant application potentials in real-world scenarios such as drug discoveries. However, existing generative models fail to achieve text-to-graph generation in the following two aspects: i) language model-based generative models struggle with generating complex graph structures, and ii) graph-based generative models mainly focus on unconditional graph generation or conditional generation with simple conditions, falling short in understanding as well as following human instructions. In this paper, we tackle the text-to-graph generation problem by employing graph diffusion models with guidance from large language models (LLMs) for the first time, to the best of our knowledge. The problem is highly non-trivial with the following challenges: 1) How to align LLMs for understanding the irregular graph structures and the graph properties hidden in human instructions, 2) How to align graph diffusion models for following natural language instructions in order to generate graphs with expected relational semantics from human. To address these challenges, we propose a novel LLM-aligned Graph Diffusion Model (**LLM-GDM**), which is able to generate graphs based on natural language instructions. In particular, we first propose the self-supervised text-graph alignment to empower LLMs with the ability to accurately understand graph structures and properties by finetuning LLMs with several specially designed alignment tasks involving various graph components such as nodes, edges, and subgraphs. Then, we propose a structure-aware cross-attention mechanism guiding the diffusion model to follow human instructions through inherently capturing the relational semantics among texts and structures. Extensive experiments on both synthetic and real-world molecular datasets demonstrate the effectiveness of our proposed **LLM-GDM** model over existing baseline methods.

## 1 Introduction

Graph generation is widely adopted in many real-world applications, such as molecule design (Xu et al., 2023; Gruver et al., 2023), social network analysis (Grover et al., 2019), code completion (Brockschmidt et al., 2019), neural architecture search (NAS) (Lee et al., 2021), *etc.*, yet its accessibility remains limited for users unfamiliar with graph concepts or coding skills since the models require expert knowledge to interact. Text-to-graph generation, that is to generate graphs following natural language instructions, holds significant application potentials in real-world scenarios. In molecular design, for instance, a researcher might specify a molecule as "soluble in water, stable at high temperatures, and effective against a specific enzyme." A text-to-graph generation model would interpret these instructions to create a graph, with nodes representing atoms and edges symbolizing bonds, thereby constructing a molecule that potentially fulfills all specified conditions. The development of such text-to-graph generation technologies can significantly accelerate the discovery and optimization of new molecules (Bhowmik et al., 2024; Moret et al., 2023; Flam-Shepherd et al., 2022; Hsu et al., 2022).

With the recent rise of large language models (LLMs), researchers (Edwards et al., 2022; Christofidellis et al., 2023; Fang et al., 2024) have begun exploring their application to text-to-graph generation. As illustrated in Figure 1, these models generate graphs by describing nodes and edges in

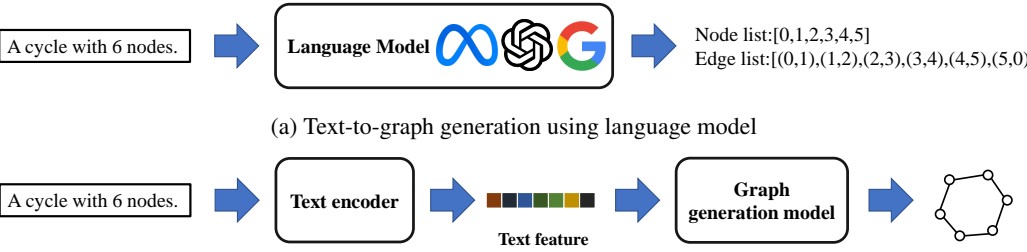

(a) Text-to-graph generation using language model

(b) Text-to-graph generation using graph generation model

Figure 1: A comparison between text-to-graph generation method using language model and graph generation model. Language models generate graphs by describing nodes and edges in textual form. The graph generation model directly generate graphs under the guidance of the text features extracted by the text encoder.

textual form. However, while text is inherently sequential, graphs exhibit more complex topological structures that capture diverse relationships between entities. This discrepancy makes it challenging for text-only models to fully grasp and generate intricate graph structures. Consequently, we argue that integrating graph-based models is crucial to advancing the task of text-to-graph generation.

Diffusion models are a class of generative models that have garnered considerable attention recently. Notably, diffusion models have been increasingly utilized in graph generation (Vignac et al., 2023; Jo et al., 2022; Kong et al., 2023) to comprehend graph structures and generate diverse graphs through learning and sampling from a given data distribution. However, their current capabilities are limited to unconditional graph generation or conditional generation with simple conditions, where the generation distribution can not be controlled by complex conditions like human natural language instructions, textual descriptions of the graph properties, *etc.*, limiting their applications in real-world scenarios.

In this paper, to bridge the gap, we study the problem of text-to-graph generation via guiding graph diffusion with large language models (LLMs), which remains unexplored in the literature. The problem is highly non-trivial with the following challenges:

- How to align large language models to understand the *irregular* graph structures and the *implied* graph properties in human instructions, where the instructions over graphs could be ambiguous.

- How to align graph diffusion models to follow natural language instructions to generate graphs with expected *relational* semantics, where nodes are inter-dependent with edges on graphs.

To address these challenges, we propose a novel LLM-aligned Graph Diffusion Model (**LLM-GDM**), which is able to generate graphs based on natural language instructions. Specifically, we first propose *self-supervised text-graph alignment* to empower LLM with better understanding of graph structures and properties by finetuning LLMs with several specially designed alignment tasks from the levels of nodes, edges and subgraphs. The finetuned LLM can extract graph-aligned features from text descriptions that capture implied graph structures in the text. Additionally, we propose a *structure-aware cross-attention mechanism* to guide the diffusion model to follow the human instructions by inherently capturing the relational semantics among texts and structures. It allows the model to generate diverse graphs that are consistent with the text input. Extensive experiments on synthetic and molecular datasets demonstrate the effectiveness of our proposed method. The contributions of this paper are summarized as follows:

- We study the problem of text-to-graph generation via guiding graph diffusion with large language models (LLMs), for the first time, to the best of our knowledge.

- We propose a novel LLM-aligned Graph Diffusion Model to generate graphs based on natural language instructions, which include two modules: i) self-supervised text-graph alignment to empower LLM with better understanding of graph structures and properties; ii) structure-aware cross-attention mechanism to capture the relational semantics among texts and structures.

- Extensive experiments on synthetic and molecular datasets demonstrate the effectiveness of our proposed method. The detailed ablation studies verify the effectiveness of each module.

## 2 PRELIMINARIES

### 2.1 GRAPH DIFFUSION MODEL

Diffusion models are a class of generative models that recently gained popularity for their excellent performance in computer vision. Recently, several works have successfully applied diffusion models to graph generation. In this paper, we focus on graph diffusion models based on stochastic differential equations (SDEs), and provide a brief description of them below.

A graph diffusion model consists of a forward process and a reverse process, both of which are defined as SDEs that operate on graph data. Consider a graph $G = (X, E)$, where $X$ represents the node features and $E$ represents the edge features. The forward process introduces Gaussian noise into $G$ as the time variable $t$ increases from $t = 0$ to $t = T$:

$$\mathrm{d}G = f(G, t)\mathrm{d}t + g(t)\mathrm{d}w, \tag{1}$$

where $f(G, t)$ and $g(t)$ are predetermined functions, and $\mathrm{d}w$ is a standard Wiener process. We denote the value of $G$ at time $t$ as $G(t)$. With appropriate choices of $f$ and $g$, $G(T)$ becomes indistinguishable from Gaussian-distributed values and contains nearly no information about the original graph $G$.

The reverse process goes backwards in time and describes the reverse of the forward SDE. It takes the following form as demonstrated in earlier works (Song et al., 2021).

$$\mathrm{d}G = [f(G, t)\mathrm{d}t - g(t)^2 \nabla_G \log p_t(G)]\mathrm{d}t + g(t)\mathrm{d}w, \tag{2}$$

where $p_t(G)$ is the marginal distribution of $G(t)$, and $\nabla_G p_t(G)$ is called its *score*.

Since the score is intractable, diffusion models use neural networks to approximate it as $s_\theta(G, t) \approx \nabla_G p_t(G)$, which can be trained using denoising score matching as follows:

$$\mathcal{L}_{\mathrm{score}} = \mathbb{E}_t \mathbb{E}_{G(0)} \mathbb{E}_{G(t)|G(0)} \left[ \lambda(t) \| s_\theta(G(t), t) - \nabla_{G(t)} \log p_{0t}(G(t) \mid G(0)) \|^2 \right], \tag{3}$$

where $p_{0t}(G(t) \mid G(0))$ is the transition probability of the forward process, and $\lambda(t)$ is a weighting function. After training, new graphs can be generated with graph diffusion models by starting with $G(T)$ sampled from Gaussian distributions and solving the reverse process using methods like Euler-Maruyama or Predictor-Corrector methods (Song et al., 2021).

### 2.2 CLASSIFIER-FREE DIFFUSION GUIDANCE

Classifier-free guidance (Ho & Salimans, 2021) is a widely used technique for conditional generation using diffusion models. It augments the score prediction network in diffusion models with the conditional information as an additional input. Let $c$ be the condition, the conditional score network $s_\theta(G, t, c)$ is trained to approximate the conditional score $\nabla_G p_t(G \mid c)$.

To improve the quality of conditional generation, classifier-free guidance introduces a scale factor that controls the influence of the condition. Let $k$ be the scale factor, classifier-free guidance modifies the estimated score function in the reverse process by scaling the difference between the predicted conditional and unconditional scores:

$$s_{\mathrm{cfg}}(G, t, c) = s_\theta(G, t) + k(s_\theta(G, t, c) - s_\theta(G, t)). \tag{4}$$

In practice, the scale factor is large than 1, so the influence of conditional data is amplified. Data generated with higher scale factors tend to be more consistent with the provided condition, while lower scale factors can result in increased generation diversity.

## 3 METHOD

In this section, we present our text-to-graph generation method. We first describe the overall framework of the proposed method, and then introduce the self-supervised text-graph alignment task for finetuning LLMs and the structure-aware cross-attention mechanism for incorporating text features into graph diffusion models.

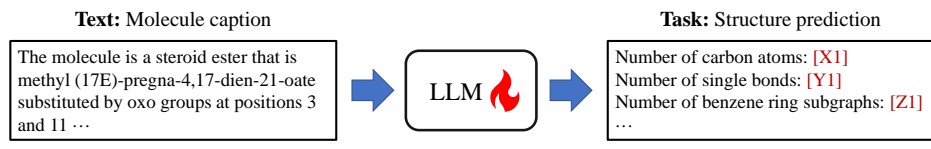

Stage 1: Finetune LLM using self-supervised text-graph alignment task

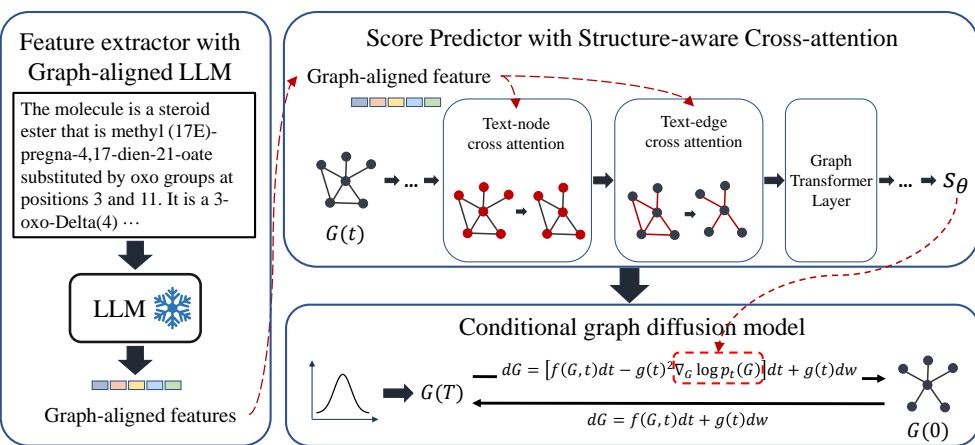

Stage 2: Use graph-aligned features to train conditional graph diffusion model

Figure 2: An overview of our method. We use a condition graph diffusion model to perform text-to-graph generation. In the first stage, as shown in the upper part of the figure, we use *self-supervised text-graph alignment* task to finetune LLM to extract graph-aligned features from the text description. In the second stage, we use the graph-aligned LLM to extract graph-aligned features from the text description, and apply them to the score predictor in the conditional graph diffusion model using *structure-aware cross-attention mechanism*.

## 3.1 FRAMEWORK

We use a diffusion model for graph generation in our method. We represent a graph $G$ by its node types $X = \{x_i\}$ and adjacency matrix $E = \{e_{ij}\}$. Here, $x_i \in \{1, 2, \ldots, C_{\text{node}}\}$ is the type of the $i$-th node in $G$, and $e_{ij} \in \{0, 12, \ldots, C_{\text{edge}}\}$ is the type of edge between nodes $i$ and $j$, with $e_{ij} = 0$ indicating no edge. Our method aims to generate corresponding graph data $G$ for a given text description $T$ by learning the conditional distribution $p_\theta(G|T)$.

The framework of our method is illustrated in Figure 2. In the first stage of our method, we use the self-supervised text-to-graph alignment task to finetune the Llama-3-8B model, obtaining a graph-aligned LLM. In the second stage, we construct a conditional graph diffusion model to generate graphs according to text description, and use a graph transformer (Yun et al., 2019) with structure-aware cross-attention as its conditional score predictor.

For a given text description $T$, the conditional score predictor uses a feature extractor with a graph-aligned LLM to extract text features from $T$, as introduced in Section 3.2. Then, the score predictor uses structure-aware cross-attention to modify the prediction results based on the extracted text features, as detailed in Section 3.3. During the generation process, we use classifier-free guidance to generate graphs according to text description $T$.

## 3.2 SELF-SUPERVISED TEXT-GRAPH ALIGNMENT

To incorporate text descriptions into the generation process, we need to extract features from the text. Pretrained language models are known to produce high-quality text features for downstream tasks. With billions of parameters, LLMs demonstrate strong performance in various graph tasks (Li et al., 2023), indicating their ability to understand the graph structure contained in text. This makes it feasible to use LLMs to extract text features relevant for graph tasks.

To ensure that the text features extracted by LLMs are more focused on the graph generation task, we finetune LLMs to obtain text features that are more relevant to the graph structure. For this purpose, we design a graph structure prediction task to finetune LLMs. Specifically, we aim for the extracted text features to reflect the structure of the graphs corresponding to the text description, including information about nodes, edge, and subgraphs. Therefore, the graph structure prediction task includes the prediction of three categories of targets: the number of nodes, edges, and subgraphs of various types in the graph, denoted by $c_{\text{node},i}$, $c_{\text{edge},i}$ and $c_{\text{sub},i}$ respectively. They are defined as follows:

$$
\begin{aligned}
c_{\text{node},i} &= |\{j \mid x_j = i\}|, \\
c_{\text{edge},i} &= |\{(j,k) \mid e_{jk} = i\}|, \\
c_{\text{sub},i} &= |\{G' \mid G' \text{ is a subgraph of } G \text{ and } G' \text{ is isomorphic to } G_i\}|.
\end{aligned}
\tag{5}
$$

The objective function for finetuning is:

$$
\mathcal{L}_{\text{align}} = \sum_i (c^*_{\text{node},i} - c_{\text{node},i})^2 + \sum_i (c^*_{\text{edge},i} - c_{\text{edge},i})^2 + \sum_i (c^*_{\text{sub},i} - c_{\text{sub},i})^2,
\tag{6}
$$

where $c^*_{\text{node},i}$, $c^*_{\text{edge},i}$, $c^*_{\text{sub},i}$ are the value predicted by LLM for the number of nodes, edges, subgraphs of various types in the graph, respectively.

It is worth mentioning that in the actual generation process, for each piece of text description, we input it into the finetuned LLM and use the output of the last hidden layer as the extracted features. The result will have features for each token in the input text.

### 3.3 STRUCTURE-AWARE CROSS-ATTENTION MECHANISM

To allow text-to-graph generation using diffusion models, it is necessary to incorporating text features into the conditional score predictor. While there are methods like affine conditioning or cross-attention for constructing the score predictors of conditional diffusion models, these approaches do not account for the complex nature of graph data. Directly applying them to nodes and edges in graph diffusion models can lead to suboptimal results. Since the edges in graphs represent the relationship between nodes, an effective conditioning method for graphs should respect this property and ensure the edge conditioning accounts for the nodes. Additionally, the size of the adjacency matrix is quadratic in the number of nodes. For more computationally expensive conditioning methods like cross-attention, calculating edge conditioning for each edge independently is costly.

In this section, we propose a structure-aware cross-attention method for graph diffusion models, which integrates a sequence of conditional text features into the score predictor network by computing attention between the graph data and the text features.

Specifically, we use the structure-aware cross-attention mechanism in the score predictor module of the graph diffusion model, where the results of node attention and edge attention are added into the network using residual connections. Let $X$ and $E$ be the node and edge features in some layer of the score predictor, and $C$ be the sequence of text features. Structure-aware cross-attention first computes the cross-attention between node features and text features as follows, and use the attention results for node conditioning:

$$
Q = X W_Q, \quad K = C W_K, \quad V = C W_V,
\tag{7}
$$

$$
A = \text{softmax}(\frac{QK^T}{\sqrt{d}}),
\tag{8}
$$

$$
X_{\text{cond}} = AV,
\tag{9}
$$

where $Q$, $K$, and $V$ are weight matrices for queries, keys, and values, $d$ is the dimensionality of keys, $A$ is the attention score for nodes, and $X_{\text{cond}}$ is the node conditioning result.

Then, structure-aware cross-attention computes the attention scores for edges based on the node attention results. For each edge $(u, v)$, its attention score should be related to the attention scores of node $u$ and $v$. We compute two scalar values $G_{1,uv}$ using a gating mechanism for each edge based on its features $E_{uv}$, which represents the influence of two endpoints $u$ and $v$ in the edge:

$$
G_{1,uv} = \sigma(E_{uv}^T W_{G1}), G_{2,uv} = 1 - G_{1,uv},
\tag{10}
$$

where $W_{G1}$ is trainable weights, and $\sigma$ is the logistic sigmoid function. The attention score of edge $(u, v)$ is computed as a weighted mixture of the attention scores for node $u$ and $v$:

$$A_{\text{edge},uv} = G_{1,uv} A_u + G_{2,uv} A_v, \tag{11}$$

where $A_{\text{edge},uv}$ is the attention score for edge $(u, v)$, and $A_u$ and $A_v$ are attention scores for node $u$ and $v$ respectively.

Finally, the edge conditioning result is determined by mixing the attention values according to the edge attention scores:

$$E_{\text{cond},uv} = A_{\text{edge},uv} V. \tag{12}$$

By deriving the edge conditioning from the node conditioning, structure-aware cross-attention can utilize the text features in the conditional score predictor with relatively low computational costs, and captures the relational semantics among texts and structures more efficiently. In our experiments, we apply structure-aware cross-attention in each layer of the score predictor.

## 4 EXPERIMENT

To demonstrate the effectiveness of our method, we constructed three parts of experiments. In the first part, we compare language model-based and graph-based methods on a synthetic graph dataset, which includes graphs with multiple types of structures. In the second part, we compare various methods on the task of text-conditional molecular graph generation using a real-world molecular dataset. In the third part, we conduct extensive ablation experiments to explore the roles of the two modules we proposed and the impact of hyper-parameter settings on the performance of text-to-graph generation.

### 4.1 GRAPH GENERATION ON SYNTHETIC DATASET

We conducted experiments on synthetic dataset, using the model to generate corresponding graphs according to the rules described in the text. We explored the ability of LLMs to understand graph structures through rules of different difficulty levels, as well as the ability of our method to understand graph structures described in the text.

**Datasets**  We construct a synthetic dataset with 7 kinds of graphs, Tree, Cycle, Wheel, Bipartite, K-regular, Component, and Mix. The details of these kinds of graphs are available in the appendix. Each graph is accompanied by its corresponding text description, which specifies its kind and important properties, like "A tree with 8 nodes" or "A graph with 10 nodes and 2 connected components". There are 10000 graphs in the training set and 1000 graphs in the test set. During testing, we ask the model to generate a new graph that matches the text description.

**Baselines**  We compare our method with several LLMs, including Qwen2.5-7B, Qwen2.5-72B, Gemma-2-9B, Gemma-2-27B, Llama-3.1-8B, and Llama-3.1-70B. We use the LLMs in a zero-shot way, asking the model to generate the text representation of graphs according to text descriptions with a prompt.

**Metrics**  For each kind of graph in the test set, we report the proportion of generated graphs that matches the given text descriptions. It is worth noting that there are generally more than one graph matching a piece of text description, and any matching graph will count towards the metric.

**Results and analysis**  The experimental results are shown in Table 1. We can find that: 1) Overall, LLMs with a larger number of parameters perform better than those with a smaller number of parameters. Moreover, LLMs other than Llama-3.1-8B perform well for generating graphs with simple structures such as trees and cycles, but perform poorly on tasks with more complex graph structures. This indicates that **LLMs struggle with generating complex graph structures**. 2) In addition, our method performs well on various tasks, indicating that **graph-based methods can better generate graphs with complex structures**.

Table 1: The result of graph generation on the synthetic dataset. The values in the table are the proportions of generated graphs that matches the given text descriptions.

| Model | Tree | Cycle | Wheel | Bipartite | K-regular | Component | Mix |
|---|---|---|---|---|---|---|---|
| Qwen2.5-7B | $0.778 \pm 0.010$ | $1.000 \pm 0.000$ | $0.020 \pm 0.016$ | $0.088 \pm 0.038$ | $0.295 \pm 0.046$ | $0.178 \pm 0.078$ | $0.178 \pm 0.019$ |
| Qwen2.5-72B | $1.000 \pm 0.000$ | $1.000 \pm 0.000$ | $0.259 \pm 0.061$ | $0.371 \pm 0.047$ | $0.767 \pm 0.017$ | $0.448 \pm 0.060$ | $0.319 \pm 0.021$ |
| Gemma-2-9B | $0.942 \pm 0.018$ | $1.000 \pm 0.000$ | $0.000 \pm 0.000$ | $0.040 \pm 0.014$ | $0.286 \pm 0.024$ | $0.146 \pm 0.005$ | $0.235 \pm 0.025$ |
| Gemma-2-27B | $1.000 \pm 0.000$ | $1.000 \pm 0.000$ | $0.000 \pm 0.000$ | $0.383 \pm 0.037$ | $0.020 \pm 0.017$ | $0.098 \pm 0.006$ | $0.231 \pm 0.048$ |
| Llama-3.1-8B | $0.178 \pm 0.037$ | $0.136 \pm 0.042$ | $0.000 \pm 0.000$ | $0.000 \pm 0.000$ | $0.007 \pm 0.010$ | $0.007 \pm 0.010$ | $0.121 \pm 0.006$ |
| Llama-3.1-70B | $1.000 \pm 0.000$ | $1.000 \pm 0.000$ | $0.597 \pm 0.069$ | $0.154 \pm 0.006$ | $0.353 \pm 0.037$ | $0.450 \pm 0.026$ | $0.347 \pm 0.067$ |
| Ours | $0.992 \pm 0.012$ | $0.636 \pm 0.018$ | $\mathbf{0.669 \pm 0.029}$ | $\mathbf{0.916 \pm 0.002}$ | $\mathbf{1.000 \pm 0.000}$ | $\mathbf{0.962 \pm 0.018}$ | $\mathbf{0.589 \pm 0.068}$ |

## 4.2 TEXT CONDITIONAL MOLECULAR GRAPH GENERATION

We conducted experiments on the text to molecule dataset and demonstrated the effectiveness of our method in generating molecular graphs based on text descriptions through comparison with the text to molecule text model.

**Datasets** Generating molecules according to text is an emerging task, and currently the only widely used and public dataset in this task is the ChEBI-20 dataset (Edwards et al., 2021). This dataset contains 33010 molecules and their corresponding text descriptions. We follow the approach of MolT5 (Edwards et al., 2022) for dataset splits and preprocessing.

**Baselines** We compare with the following methods.

- RNN: A 4-layer GRU recurrent neural network with bidirectional encoder trained on the ChEBI-20 dataset.
- Transformer: A vanilla transformer model consisting of six encoder and decoder layers trained on the ChEBI-20 dataset.
- MolT5: An encoder-decoder Transformer model initialized with a public checkpoint of T5, then pretrained on the combined dataset of C4 and ZINC, finally finetuned on the ChEBI-20 dataset.
- Llama-3.1: An decoder-only Transformer model published by Meta. Llama-3.1-8B has 8B parameters, while Llama-3.1-70B has 70B parameters. We use the model in a zero-shot way, asking the model to generate the SMILES representation of molecules according to text descriptions with a prompt.

**Metrics** We use the following metrics to evaluate our method and compare it with the baseline.

- Fingerprint-based molecule similarity metrics: We measure the fingerprint Tanimoto similarity (FTS) between each generated molecule and the corresponding ground truth molecule. MACCS, RDK, and Morgan represent the three different extraction methods for molecular fingerprints. We consider these to be the primary metrics measuring the quality of generated graphs in our experiments.
- FCD: We measure the Fréchet ChemNet Distance (FCD) (Preuer et al., 2018) between the generated molecules and ground truth molecules. It reflects the distance of chemical and biological information between the two sets of molecules.
- Exact: We measure the proportion of generated molecules that are exact matches of their corresponding ground truth molecules.
- Validity: We report the validity of the generated molecules as measured by RDKit sanitization.
- Diversity: We measure the diversity of generated molecules, defined as the average pairwise differences between multiple molecules generated by the model guided by the same text description. The pairwise differences are calculated by subtracting their FTS from 1. In the experiments, we select the first 100 text descriptions in the test set and generate 10 graphs for each description to calculate the diversity metrics.

**Results and analysis** From Table 2, we can observe that: 1) our method has similar performance to MolT5 in terms of FTS metrics, outperforming MolT5 in MACCS and RDK FTS, and has much smaller trainable parameter sizes. 2) Our method generates relatively few exact molecule matches,

Table 2: The result of molecule generation guided by the text in the test split of CheBI-20. "MACCS", "RDK", and "Morgan" are fingerprint Tanimoto similarity metrics. The result of RNN, Transformer, and MolT5 are sourced from Edwards et al. (2022), and other methods maintained the same settings as Edwards et al. (2022) during testing. "Param." denote the number of trainable parameters of the model. The number of parameters for RNN and Transformer are unknown.

| Model | MACCS ↑ | RDK ↑ | Morgan ↑ | FCD↓ | Exact ↑ | Validity↑ | Param. |
|---|---|---|---|---|---|---|---|
| RNN | 0.591 | 0.400 | 0.362 | 4.55 | 0.005 | 0.542 | — |
| Transformer | 0.480 | 0.320 | 0.217 | 11.32 | 0.000 | 0.906 | — |
| MolT5 | 0.721 | 0.588 | **0.529** | **2.18** | **0.081** | 0.772 | 220M |
| Llama-3.1-8B | $0.545 \pm 0.004$ | $0.305 \pm 0.004$ | $0.238 \pm 0.002$ | $5.86 \pm 0.15$ | $0.007 \pm 0.001$ | $0.370 \pm 0.005$ | — |
| Llama-3.1-70B | $0.683 \pm 0.003$ | $0.450 \pm 0.004$ | $0.390 \pm 0.004$ | $3.27 \pm 0.10$ | $0.049 \pm 0.003$ | $0.563 \pm 0.008$ | — |
| Ours | $\mathbf{0.787 \pm 0.001}$ | $\mathbf{0.638 \pm 0.002}$ | $0.470 \pm 0.002$ | $2.96 \pm 0.06$ | $0.050 \pm 0.002$ | $\mathbf{1.000 \pm 0.000}$ | 16.5M |

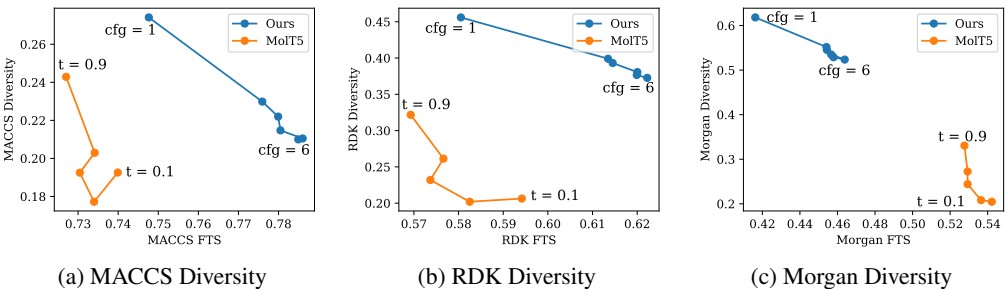

(a) MACCS Diversity  (b) RDK Diversity  (c) Morgan Diversity

Figure 3: The diversity results of our method and MolT5. The horizontal axis represents the FTS between the generated and real molecules, the vertical axis represents the diversity between the generated molecules. The orange dots represent the values measured by MolT5 when setting different temperature values, and the blue dots represent the values measured by our method when setting different classifier-free guidance scale values.

because the generated graphs of diffusion models are diverse and it is difficult to ensure full matches. 3) At the same time, we can also observe that zero-shot Llama-3.1 performs poorly on various metrics, indicating that directly using LLMs for molecular generation tasks does not result in good performances.

In addition, we report the diversity results of our method and MolT5. Since diversity can be affected by the choice of parameters like the temperature value for MolT5 and the classifier-free guidance scale for our method, we test the models using several configurations and report their FTS and diversity metrics. The results can be seen in Figure 3. It can be seen that 1) the FTS metrics generally increases when the temperature is reduced (for MolT5) or the CFG scale is increased (for our method), while the diversity metrics generally decreases. 2) Overall, our method has higher diversity scores than MolT5. When the two methods have similar FTS metrics, our method achieves better diversity. While our method has lower performance than MolT5 in the Morgan FTS metric, the diversity of our method is significantly higher. The result demonstrates that our proposed method can generate graphs with higher diversity for the task of text-to-graph generation.

### 4.3 ABLATION STUDY

We conducted ablation experiments to explore the effects of different settings. All ablation experiments are performed on the ChEBI-20 dataset. The metrics are the same as the molecule generation experiments in Section 4.2.

#### 4.3.1 THE EFFECT OF DIFFERENT TEXT CONDITIONING METHODS

We compare our structure-aware cross-attention mechanism with other text conditioning methods for diffusion models.

**Compared methods** We compare the following methods:

Table 3: Molecule generation results of different text conditioning methods.

| Model | MACCS ↑ | RDK ↑ | Morgan ↑ | FCD↓ | Exact ↑ |
|---|---|---|---|---|---|
| Affine | 0.734 | 0.570 | 0.391 | 3.67 | 0.028 |
| Cross-attention | 0.770 | 0.615 | 0.440 | 3.28 | 0.040 |
| Ours | **0.789** | **0.639** | **0.473** | **2.98** | **0.048** |

Table 4: Molecule generation results of LLM finetuning.

| Model | MACCS ↑ | RDK ↑ | Morgan ↑ | FCD↓ | Exact ↑ |
|---|---|---|---|---|---|
| No LLM finetune | 0.728 | 0.574 | 0.385 | 4.23 | 0.031 |
| LLM finetune | 0.789 | 0.639 | 0.473 | 2.98 | 0.048 |

- Affine: The features of the last token in the text description are inserted into the score predictor using feature-wise affine transformations, also known as feature-wise linear modulation (FiLM).
- Cross-attention: The text features are inserted into the score predictor using cross-attention between node features and text features. The edge features are not modified directly.
- Ours: The text features are inserted into the score predictor using our proposed structure-aware cross-attention mechanism.

**Results and analysis**    The experimental results are shown in Table 3. We can find that structure-aware cross-attention achieves the best performance among compared methods in terms of molecular similarity to the ground truth. This indicates that our proposed method can better incorporate text features into graph diffusion models.

### 4.3.2    THE EFFECT OF LLM FINETUNING

**Compared methods**    We compare with the following experimental settings:

- No LLM finetune: The model uses pretrained Llama-3-8B directly to extract features from text.
- LLM finetune: The model uses our finetuned LLM to extract graph-aligned features from text.

**Results and analysis**    The experimental results are shown in Table 4. We can find that using the finetuned LLM for feature extraction is crucial for improving the quality of graph generation. Although the validity has slightly decreased, other metrics have significantly improved with the finetuned LLM.

### 4.3.3    THE EFFECT OF CLASSIFIER-FREE GUIDANCE SCALE

It is known that the classifier-free guidance scale has a significant impact on the performance of text-to-image diffusion models. In this section, we explore the effect of different guidance scales on our text-to-graph model.

**Results and analysis**    The experimental results are shown in Table 5. It can be seen that: 1) The validity of generated molecules decreases as the guidance scale increases, indicating that strong

Table 5: Molecule generation results using different classifier-free guidance scales.

| Model | MACCS ↑ | RDK ↑ | Morgan ↑ | FCD↓ | Exact ↑ |
|---|---|---|---|---|---|
| CFG scale = 1 | 0.752 | 0.584 | 0.411 | 3.355 | 0.037 |
| CFG scale = 3 | 0.787 | 0.635 | 0.465 | 2.98 | 0.048 |
| CFG scale = 5 | 0.789 | 0.639 | 0.473 | 2.98 | 0.048 |
| CFG scale = 7 | 0.575 | 0.432 | 0.297 | 3.005 | 0.026 |
| CFG scale = 9 | 0.336 | 0.210 | 0.111 | 2.97 | 0.002 |

guidance from the text can lead to generating inconsistent graph structures. 2) The influence of guidance scale on molecular similarity metrics are minimal, indicating that our method is robust to the choice of guidance scale. Generally, choosing a guidance scale around 5 leads to the best performance.

## 5    RELATED WORK

### 5.1    DIFFUSION-BASED GRAPH GENERATION

Diffusion model has achieved great success in the field of computer vision. Recently, some researchers have used diffusion models to solve graph generation task. For example, EDP-GNN (Niu et al., 2020) is the first work using Score Matching with Langevin Dynamics (SMLD) (Song & Ermon, 2019) diffusion model to generate graphs, which learns the score function of the adjacency matrices distributions of the graphs. GDSS (Jo et al., 2022) proposes a graph generation method using continuous-time diffusion models (Song et al., 2021), which models the joint distribution of the nodes and edges through stochastic differential equations (SDEs). DiGress (Vignac et al., 2023) uses a diffusion model over discrete data space for graph generation, and additionally preserves the marginal distribution of node and edge types and incorporates auxiliary graph-theoretic features. These methods have demonstrated excellent performance on the task of graph generation.

In order to generate graphs that match specific requirements, conditional graph generation has received attention in recent years. For example, DiGress (Vignac et al., 2023) uses classifier guidance to perform graph generation guided by several graph-level properties, like the dipole moment and highest occupied molecular orbit of molecular graphs. However, existing methods have not explored the task of text-guided graph generation, which is necessary for the popularization of graph generation methods in various fields.

### 5.2    LARGE LANGUAGE MODELS FOR GRAPHS

LLMs have achieved good results on various natural language process tasks. Recently, many works explored the application of LLMs in graph tasks. For example, Chen et al. (2023) uses LLMs to predict node categories on text attribute graphs. Wang et al. (2023) proposes a benchmark framework to evaluate the performance of LLMs with several graph algorithmic tasks, including topological sort, maximum flows, *etc*. Zhang et al. (2023) evaluates the abilities of LLMs to handle spatial-temporal information on dynamic graphs. Yao et al. (2024) designs a series of tasks to evaluate the graph generation ability of LLMs. These and other studies have demonstrated the potential of LLMs for processing graph tasks, showing the possibility to extract graph structure features from text description using LLMs.

Christofidellis et al. (2023) and Fang et al. (2024) are text-to-molecule methods that leverage language models for generation, where molecules are represented in the SMILES format. Gong et al. (2024) employs a text diffusion model to generate the SMILES representation of molecules. While these methods focus on generating molecules from text representations, they do not extend to other graph generation tasks. Zhu et al. (2024) utilizes a latent space diffusion model to generate latent features, which are then decoded into molecule graphs using a graph decoder. In contrast, this paper explores for the first time the use of graph-aligned LLMs to extract text features for guiding conditional graph diffusion models.

## 6    CONCLUSION

This paper addresses the critical gap in text-to-graph generation by proposing the LLM-aligned Graph Diffusion Model (LLM-GDM), which integrates large language models with graph diffusion model to generate graphs from natural language instructions. By developing a self-supervised text-graph alignment process and introducing a structure-aware cross-attention mechanism, our approach enhances the model's understanding of graph structures and properties and ensures that generated graphs adhere to the specified relational semantics in the text. Extensive experimental results on synthetic and molecular datasets confirm the efficiency of our method.

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

## A    EXPERIMENTAL SETTING

Our code is based on GDSS-Transformer (https://github.com/DongkiKim95/GDSS-Transformer/). We modified the code to allow text-conditional graph generation. We initialized the parameters of the base architecture with the checkpoint of GDSS-Transformers's model trained on ZINC250k, and then trained our structure-aware cross-attention along with the base architecture with a learning rate of 2e-4 on ChEBI-20. For LLM finetuning, we use LoRA with rank set to 8 and alpha set to 64, and trained for 6 epochs.

All experiments are performed with a single NVIDIA A100-SXM4-40GB. Finetuning the LLM takes about 5 hours, and training our diffusion model on ChEBI-20 takes about 1 day per run. Evaluating the diffusion model on ChEBI-20 takes about 4 hours per run.

## B    DATASETS

The synthetic dataset we used in the experiment was constructed based on the following rules:

- **Tree**: A tree with the specified number of node. An example of text description in the dataset is "A tree with 12 nodes".
- **Cycle**: A cycle with the specified number of nodes. An example of text description in the dataset is "A tree with 12 nodes".
- **Wheel**: A graph formed by connecting a single node to all nodes of a cycle. An example of text description in the dataset is "A wheel with 12 nodes".
- **Bipartite**: A complete bipartite graph with the specified number of nodes. An example of text description in the dataset is "A complete bipartite graph with 6 nodes and 8 nodes in each split".
- $K$-**regular graphs**: A graph whose every node have the same number of neighbors. An example of text description in the dataset is "A 2-regular graph with 12 nodes".
- **Component**: A graph with the specified number of nodes and connected components. An example of text description in the dataset is "A graph with 12 nodes and 3 connected components".
- **Mix**: A graph with the specified number of nodes and connected components, and every component is a graph that satisfies a certain rule. An example of text description in the dataset is "A graph with 12 nodes and 3 components. 2 components are trees. 1 component is a cycle.".

## C    COMPUTATIONAL COSTS

Compared to unconditional graph diffusion models, our method only requires a small increase in training and inference time. For training our model, our method has the additional step of LLM finetuning, which takes a few days but only needs to be done once. For inference, our method, with the addition of text feature extraction and structure-aware cross-attention, only increased the inference time by about 30%.

## D    MORE EXPERIMENT RESULTS

We list the numerical results of the diversity metrics in Table 6 in complement to the visualization in Figure 3.

In order to better illustrate the quality of generated graphs of our method, we list in Table 7 some comparison results with MolT5. It can be seen from the table that our method can generate diverse results while maintaining the basic molecular structure, while the diversity and accuracy of MolT5 is generally worse. This indicates that our method has more potential in the application of text-to-graph generation.

Table 6: The values of diversity and FTS under different parameters when generating molecules using various methods. "MACCS", "RDK", "Morgan" are three types methods calculating molecule fingerprint. "Diver." is diversity metric value. "FTS" is fingerprint-based molecule similarity metric.

| Model | MACCS Diver. | MACCS FTS | RDK Diver. | RDK FTS | Morgan Diver. | Morgan FTS |
|-------|-------------|-----------|------------|---------|---------------|------------|
| Ours (CFG = 1) | 0.748 | 0.726 | 0.581 | 0.544 | 0.416 | 0.382 |
| Ours (CFG = 2) | 0.776 | 0.770 | 0.613 | 0.601 | 0.454 | 0.448 |
| Ours (CFG = 3) | 0.780 | 0.778 | 0.615 | 0.607 | 0.454 | 0.455 |
| Ours (CFG = 4) | 0.780 | 0.785 | 0.620 | 0.619 | 0.457 | 0.465 |
| Ours (CFG = 5) | 0.786 | 0.790 | 0.620 | 0.623 | 0.464 | 0.476 |
| Ours (CFG = 6) | 0.785 | 0.790 | 0.622 | 0.627 | 0.458 | 0.471 |
| Ours (CFG = 7) | 0.474 | 0.543 | 0.332 | 0.397 | 0.213 | 0.275 |
| Ours (CFG = 8) | 0.321 | 0.398 | 0.195 | 0.264 | 0.099 | 0.158 |
| Ours (CFG = 9) | 0.314 | 0.374 | 0.191 | 0.243 | 0.098 | 0.140 |
| Ours (CFG = 10) | 0.359 | 0.338 | 0.230 | 0.217 | 0.137 | 0.119 |
| MolT5 (t = 0.1) | 0.740 | 0.807 | 0.594 | 0.794 | 0.542 | 0.795 |
| MolT5 (t = 0.2) | 0.746 | 0.807 | 0.596 | 0.784 | 0.543 | 0.784 |
| MolT5 (t = 0.3) | 0.734 | 0.823 | 0.583 | 0.798 | 0.536 | 0.792 |
| MolT5 (t = 0.4) | 0.740 | 0.821 | 0.585 | 0.786 | 0.538 | 0.777 |
| MolT5 (t = 0.5) | 0.730 | 0.807 | 0.574 | 0.768 | 0.529 | 0.756 |
| MolT5 (t = 0.7) | 0.736 | 0.807 | 0.580 | 0.757 | 0.536 | 0.744 |
| MolT5 (t = 0.8) | 0.734 | 0.797 | 0.577 | 0.739 | 0.529 | 0.727 |
| MolT5 (t = 0.9) | 0.734 | 0.798 | 0.574 | 0.740 | 0.525 | 0.719 |
| MolT5 (t = 1.0) | 0.727 | 0.757 | 0.569 | 0.678 | 0.528 | 0.669 |

Table 7: Visualization of generated molecules by our method and MolT5.

| Text description | Ground truth | Our method | MolT5 |
|------------------|--------------|------------|-------|
| The molecule is an indolylmethylglucosinolate that is the conjugate base of 4-methoxyglucobrassicin, obtained by deprotonation of the sulfo group. It is a conjugate base of a 4-methoxyglucobrassicin. |  |  |  |
| The molecule is a member of the class of naphthoates that is 1-naphthoate substituted at positions 3 and 5 by hydroxy and methyl groups respectively; major species at pH 7.3. It has a role as a bacterial metabolite. It is a conjugate base of a 3-hydroxy-5-methyl-1-naphthoic acid. |  |  |  |
| The molecule is a myricetin O-glucuronide that is myricetin with a beta-D-glucosiduronic acid residue attached at the 5-position. It has a role as a metabolite. It is a myricetin O-glucuronide, a pentahydroxyflavone, a member of flavonols and a monosaccharide derivative. |  |  |  |