# OpenReview forum: "Text-to-graph Generation with Conditional Diffusion Models Guided by Graph-aligned LLMs"
_ICLR.cc/2025/Conference — Submitted to ICLR 2025_

### Official Review · Reviewer_yNGe · 2024-10-30

**Soundness:** 1
**Presentation:** 3
**Contribution:** 2
**Rating:** 3
**Confidence:** 4

**Summary:**

The paper focuses on large language model (LLM)-guided diffusion models for graph generation tasks. It first fine-tunes the LLM with a self-supervised task of counting the number of nodes, edges, and subgraphs. Then, it uses the output from the last layer of the LLM as the feature vector for input texts in the diffusion model, where this feature vector guides the generation process. Experiments on synthetic datasets and molecular graphs demonstrate the promise of the proposed method. The work is interesting and could be improved with clearer motivation, refined method design, and enhanced experiments.

**Strengths:**

- The paper is easy to follow.
- This paper proposes to incorporate large language models (LLMs) and graph diffusion models to tackle the challenge of generating complex graphs for LLMs. By using fine-tuned LLMs to generate informative feature vectors for diffusion models, it bridges the gap between textual information and graph structure, enabling more sophisticated graph reasoning.

**Weaknesses:**

Motivation:
- The claim (lines 17-18, 77-80) that “Graph-based generative models mainly focus on unconditional graph generation, falling short in understanding as well as following human instructions” is inaccurate. Graph diffusion models have rapidly evolved for conditional generation. The paper mentions two graph diffusion models, GDSS and DiGress, both of which have conditional versions developed in [1,2]. The authors need a more comprehensive survey of recent work advancing conditional graph diffusion models.
- It is unclear why LLMs are necessary to guide diffusion models. LLMs are primarily designed for generative tasks; however, this paper uses them for embedding. Although this is an interesting direction, it is unclear why a language model specialized in producing embeddings isn’t used instead. Fine-tuning LLMs requires substantial resources, and the motivation for using LLMs in this context is not clearly established.

Method: Graphs have discrete structures, and DiGress has been proposed to enhance GDSS for improved graph modeling. It is unclear whether the proposed method uses Gaussian noise instead of the discrete noise used in DiGress. Additionally, many self-supervised tasks have been proposed in graph learning, but the rationale for the one chosen for text-graph alignment is unclear. For example, why would LLMs already align with graphs simply by counting edges/nodes? This task seems too superficial for an LLM to grasp complex graph structures.

Experiments:
- The work compares the proposed method with a few LLMs but overlooks diffusion models or other generative models designed for graph generation, making the results less convincing.
- The metrics are not convincing. There is no metric reflecting the model's ability to satisfy the requirements specified in the text.
- The paper lacks a comprehensive comparison with text-to-graph generation baselines and evaluation on more domains of real-world datasets.

Reference:
- [1] DiGress: Discrete Denoising diffusion for graph generation. ICLR 2023.
- [2] Exploring Chemical Space with Score-based Out-of-distribution Generation. ICML 2023.

**Questions:**

- Please address the concerns and answer the questions listed under weaknesses.
- What large language model is used in the proposed method?
- What graph neural network architecture is used in the work?

---

> ### Author Response · Authors · 2024-11-25
>
> Thank you for your detailed review and thoughtful comments.
>
> Please find our responses to your points below:
>
> **Q:** The explanation of the claim (lines 17-18, 77-80).
>
> **A:** Thank you for pointing out the inaccuracies in our phrasing. We have updated the wording in the revised version of the paper. By "human instructions," we refer to commands expressed in natural language, such as "Generate a graph with 5 cycles."
> While some methods provide conditional generation capabilities [1,2], they lack the ability to perform text-to-graph generation. These methods rely on the gradient of a separate prediction network to guide the diffusion sampling process, which does not scale well to more complex conditions, such as text descriptions. Specifically, (1) the prediction network often struggles to produce high-quality results, and (2) the gradient surface becomes highly complex, making the sampling process challenging to converge [3].
> In contrast, our method employs a classifier-free guidance strategy, leveraging text features extracted through graph-aligned LLMs. This approach enables the model to accurately generate graphs based on complex textual instructions.
>
> [1] DiGress: Discrete Denoising diffusion for graph generation. ICLR 2023.
>
> [2] Exploring Chemical Space with Score-based Out-of-distribution Generation. ICML 2023.
>
> [3] Classifier-Free Diffusion Guidance. NIPS 2021.
>
> **Q:** Why LLMs are necessary to guide diffusion models.
>
> **A:** Generating graphs from textual descriptions is a critical application with significant implications in fields such as drug discovery, code generation, and beyond. As large language models (LLMs) are currently the most effective models for processing text, their ability to understand and interpret text is central to this task. However, existing LLMs encounter challenges when it comes to understanding graph structures.
> To address this issue, we have designed a self-supervised text-graph alignment task to fine-tune LLMs, enabling them to extract graph-related features from text. This, in turn, allows the LLM to guide the graph diffusion model in generating graphs that are well-aligned with the provided textual descriptions. The ablation study in Table 4 further confirms that our fine-tuned LLM outperforms the original LLM in guiding the graph diffusion model for generation.
>
> **Q:** Whether the proposed method uses Gaussian noise instead of the discrete noise used in DiGress.
>
> **A:** We use Gaussian noise, as it is more general and applicable to both continuous and discrete features.
>
> **Q:** Why would LLMs already align with graphs simply by counting edges/nodes?
>
> **A:** Accurately counting the different types of nodes, edges, and subgraphs is crucial for understanding graph structures [4,5]. The ability of an LLM to perform this task suggests that it has partially reconstructed the entire graph internally (though not necessarily in a graph format). This implies that the large language model can be considered aligned with the graph. Ablation studies demonstrate that our fine-tuned LLM outperforms the original LLM in guiding the graph diffusion model for generation.
>
> [4] Graph Self-Supervised Learning: A Survey. TKDE 2022.
>
> [5] Self-Supervised Learning of Graph Neural Networks: A Unified Review. TPAMI 2022.
>
> **Q:** Compare with diffusion models or other generative models designed for graph generation.
>
> **A:** Other diffusion models and graph generation models are unable to generate graphs based on text instructions, so there is no direct baseline for comparison. While [6] claims to achieve this, it has not been evaluated on widely accepted task-specific metrics (such as FTS, FCD, etc.), and the code has not been made publicly available.
>
> [6] 3M-Diffusion: Latent Multi-Modal Diffusion for Language-Guided Molecular Structure Generation
>
> **Q:** There is no metric reflecting the model's ability to satisfy the requirements specified in the text.
>
> **A:** In the synthetic dataset, we use accuracy as the metric, evaluating whether the generated graph aligns with the given text. For the molecular dataset, there isn't a direct equivalent metric. Instead, we assess the quality of the generated molecular graph by comparing various similarity measures (e.g., fingerprint-based molecule similarity metrics, FCD) between the generated graph and the real molecular graph to determine how well the generated result meets the textual requirements. Examples of molecular graphs generated from given text descriptions are provided in the appendix for reference.
>
> (to be continued in the next comment)

---

> > ### Author Response · Authors · 2024-11-25
> >
> > (continued)
> >
> > **Q:** More baselines and datasets.
> >
> > **A:**
> > - The field of text-to-graph generation remains relatively unexplored, resulting in a limited number of directly comparable baselines. For the experiments conducted on the synthetic dataset, we carefully selected several state-of-the-art LLMs as baselines to ensure a robust evaluation. For the real-world dataset, we included MolT5, a widely recognized approach for generating molecules from textual descriptions. Additionally, we compared our method with Llama3, a recently introduced general-purpose LLM that demonstrates molecule generation capabilities.
> > - ChEBI-20 is widely recognized as the leading dataset for the text-to-molecule generation task, comprising 33,010 molecule-description pairs. As highlighted in prior works [7,8,9,10], this dataset is the primary benchmark used in similar studies. Additionally, we also created a novel text-to-graph dataset focused exclusively on topological structures and conducted extensive experiments on it. Our results demonstrate that, compared to current LLMs, our method achieves superior performance in generation tasks.
> >
> > [7] Unifying Molecular and Textual Representations via Multi-task Language Modelling. ICML 2023.
> >
> > [8] Text-Guided Molecule Generation with Diffusion Language Model. AAAI 2024.
> >
> > [9] MolT5: Translation between Molecules and Natural Language. EMNLP 2022.
> >
> > [10] BioT5: Enriching Cross-modal Integration in Biology with Chemical Knowledge and Natural Language Associations. EMNLP 2023.
> >
> > **Q:** What large language model is used in the proposed method?
> >
> > **A:** In the first stage of our method, we use the self-supervised text-to-graph alignment task to finetune the ***Llama-3-8B*** model, obtaining a graph aligned LLM.
> >
> > **Q:** What graph neural network architecture is used in the work?
> >
> > **A:** In the second stage, we construct a conditional graph diffusion model to generate graphs according to text description, and use a ***graph transformer*** with structure-aware cross-attention as its conditional score predictor.

---

> ### Comment · Reviewer_yNGe · 2024-11-28
> **response after rebuttal**
>
> I acknowledge that I've read the rebuttal. I also have read other reviews. The comments / responses may help the authors improve their work. However, the responses cannot address most of my concerns. I choose to keep my evaluation.

---

### Official Review · Reviewer_qbf1 · 2024-11-03

**Soundness:** 3
**Presentation:** 3
**Contribution:** 2
**Rating:** 3
**Confidence:** 3

**Summary:**

This paper aims to deal with text-to-graph generation using graph diffusion and LLMs. Specifically, this paper proposes a  LLM-aligned graph diffusion model with two main modules to achieve this goal, self-supervised text-graph alignment and structure-aware cross-attention. Three sets of experiments on different tasks verify the effectiveness of the proposed model. Ablation analysis is also conducted.

**Strengths:**

1. This paper motivates the problem clearly with clear challenges and proposed solutions. This paper also includes a figure in the Introduction section to aid the description.

2. The Preliminaries section contains background information to make the paper self-contained. Overall, the model architecture is clearly described.

3. Experiments are relatively sufficient to verify the effectivenes of the proposed model, and ablation analysis in also conducted to show the importance of each module.

**Weaknesses:**

Though this paper uses an interesting method to solve the text-to-graph generation problem, from my point of view there are some unacceptable shortcomings.

1. When we consider the generation problem between text modality and graph modality, we should usually solve both sides of generation, i.e., text-to-graph generation and graph-to-text generation. Existing works [1] of generation between text and graph consider both sides, but the submitted paper models only text-to-graph, but not the other side of generation, which significantly limits the technical contribution.

[1] Edwards, C., Lai, T., Ros, K., Honke, G., Cho, K., & Ji, H. (2022). Translation between Molecules and Natural Language. In 2022 Conference on Empirical Methods in Natural Language Processing, EMNLP 2022.

2. When we do experiments, we encourage authors to repeat the same experiment setting multiple times and report both mean and standard deviation, or conduct significance t-test. However, this paper doesn't have standard deviation or significance t-test, which is difficult for readers to judge how significantly the proposed model outperforms baselines.

3. This paper emphasizes computational costs in the experiments, but there is a lack of computational complexity analysis.

**Questions:**

1. For Eq. 8, the standard multi-head self-attention in the Transformer paper divides the product of $ Q $ and $ K $ by the squared root of dimension $ \sqrt{d} $, not $ d $. Why do authors divide $ QK^\top $ by $ d $ here?

---

> ### Author Response · Authors · 2024-11-25
>
> Thank you for your detailed review and valuable comments.
>
> Please find our responses to your points below:
>
> **Q:** The reason why graph-to-text generation was not included in our work.
>
> **A:** In this paper, we focus on the text-to-graph generation problem, specifically leveraging a text-conditioned diffusion model for graph generation. As such, we do not address the graph-to-text generation problem, as it lies beyond the scope of our study. Diffusion-based methods are inherently one-directional, and existing diffusion-based approaches [1, 2] have exclusively explored the text-to-molecule task without addressing the molecule-to-text problem.
> Furthermore, it is important to highlight that text-to-graph generation is fundamentally different from the text-to-molecule generation task explored in [3]. We included molecule generation in our text-to-graph experiments because molecules can naturally be represented as graphs, thereby serving as a relevant test case for our method.
>
> [1] Text-Guided Molecule Generation with Diffusion Language Model. AAAI 2024.
>
> [2] 3M-Diffusion: Latent Multi-Modal Diffusion for Language-Guided Molecular Structure Generation. COLM 2024.
>
> [3] Translation between Molecules and Natural Language. In 2022 Conference on Empirical Methods in Natural Language Processing, EMNLP 2022.
>
> **Q:** Add both mean and standard deviation in the paper.
>
> **A:** Thank you for your suggestion. We have added both the mean and standard deviation in the latest version of the paper that we uploaded.
>
> ***The result of graph generation on the synthetic dataset.***
> | Model | Tree | Cycle | Wheel | Bipartite | K-regular | Component | Mix |
> |---------------|----------------------------|----------------------------|----------------------------|----------------------------|----------------------------|----------------------------|----------------------------|
> | Qwen2.5-7B | $0.778 \pm 0.010$ | $\mathbf{1.000 \pm 0.000}$ | $0.020 \pm 0.016$ | $0.088 \pm 0.038$ | $0.295 \pm 0.046$ | $0.178 \pm 0.078$ | $0.178 \pm 0.019$ |
> | Qwen2.5-72B | $\mathbf{1.000 \pm 0.000}$ | $\mathbf{1.000 \pm 0.000}$ | $0.259 \pm 0.061$ | $0.371 \pm 0.047$ | $0.767 \pm 0.017$ | $0.448 \pm 0.060$ | $0.319 \pm 0.021$ |
> | Gemma-2-9B | $0.942 \pm 0.018$ | $\mathbf{1.000 \pm 0.000}$ | $0.000 \pm 0.000$ | $0.040 \pm 0.014$ | $0.286 \pm 0.024$ | $0.146 \pm 0.005$ | $0.235 \pm 0.025$ |
> | Gemma-2-27B | $\mathbf{1.000 \pm 0.000}$ | $\mathbf{1.000 \pm 0.000}$ | $0.000 \pm 0.000$ | $0.383 \pm 0.037$ | $0.020 \pm 0.017$ | $0.098 \pm 0.006$ | $0.231 \pm 0.048$ |
> | Llama-3.1-8B | $0.178 \pm 0.037$ | $0.136 \pm 0.042$ | $0.000 \pm 0.000$ | $0.000 \pm 0.000$ | $0.007 \pm 0.010$ | $0.007 \pm 0.010$ | $0.121 \pm 0.006$ |
> | Llama-3.1-70B | $\mathbf{1.000 \pm 0.000}$ | $\mathbf{1.000 \pm 0.000}$ | $0.597 \pm 0.069$ | $0.154 \pm 0.006$ | $0.353 \pm 0.037$ | $0.450 \pm 0.026$ | $0.347 \pm 0.067$ |
> | Ours | $0.992 \pm 0.012$ | $0.636 \pm 0.018$ | $\mathbf{0.669 \pm 0.029}$ | $\mathbf{0.916 \pm 0.002}$ | $\mathbf{1.000 \pm 0.000}$ | $\mathbf{0.962 \pm 0.018}$ | $\mathbf{0.589 \pm 0.068}$ |
>
> ***The result of molecule generation guided by the text in the test split of CheBI-20.***
>
> | Model | MACCS $\uparrow$ | RDK $\uparrow$ | Morgan $\uparrow$ | FCD $\downarrow$ | Exact $\uparrow$ | Validity $\uparrow$ | Param. |
> |---------------|-----------------------------|----------------------------|------------------------------|-------------------|--------------------|----------------------------|--------|
> | Llama-3.1-8B | $0.545 \pm 0.004$ | $0.305 \pm 0.004$ | $0.238 \pm 0.002$ | $5.86 \pm 0.15$ | $0.007 \pm 0.001$ | $0.370 \pm 0.005$ | --- |
> | Llama-3.1-70B | $0.683 \pm 0.003$ | $0.450 \pm 0.004$ | $0.390 \pm 0.004$ | $3.27 \pm 0.10$ | $0.049 \pm 0.003$ | $0.563 \pm 0.008$ | --- |
> | Ours | $\mathbf{0.787 \pm 0.001}$ | $\mathbf{0.638 \pm 0.002}$ | $0.470 \pm 0.002$ | $2.96 \pm 0.06$ | $0.050 \pm 0.002$ | $\mathbf{1.000 \pm 0.000}$ | 16.5M |
>
>
> **Q:** Computational complexity analysis
>
> **A:** Our structure-aware cross-attention mechanism has the computational complexity of $O((N + L) d^2 + N^2 L d)$, while the standard cross-attention mechanism applied to both nodes and edges has the complexity of $O((N^2 + L) d^2 + N^2 L d)$, where $N$ is the number of nodes, $L$ is the number of text tokens, and $d$ is the feature dimension.
> The graph transformer architecture has the complexity $O(N^2 d^2)$ for each layer. Importantly, the complexity introduced by our structure-aware cross-attention module is non-dominating compared to the existing layers, ensuring computational efficiency.
>
> **Q:** The typo in Equation 8.
>
> **A:** Thank you for reading carefully, this was a typo and we have corrected it in the revised paper.

---

### Official Review · Reviewer_JV2T · 2024-11-03

**Soundness:** 3
**Presentation:** 2
**Contribution:** 2
**Rating:** 5
**Confidence:** 4

**Summary:**

This paper investigates the intriguing problem of text-to-graph generation, which involves creating graphs based on natural language instructions and has significant potential applications like drug discovery. The authors point out that existing diffusion-based graph generation models mainly focus on unconditional graph generation and lack the ability to comprehend and follow human instructions. To address this issue, they explore guiding graph diffusion with large language models (LLMs) for text-to-graph generation—an area that has not been thoroughly explored. Their extensive experiments on synthetic and molecular datasets demonstrate the effectiveness of their model.

**Strengths:**

1. This paper examines the intriguing problem of text-to-graph generation, which involves generating graphs based on natural language instructions.
2.  This paper has some reasonable experiments to show the effectiveness of their method.
3.  Self-supervised finetuning and structure-aware cross attention make sense in the solution.

**Weaknesses:**

1. ChEBI-20 is the only dataset on which all models were tested for real-world text-guided graph generation. I suggest that the authors include additional real-world datasets to verify the effectiveness of the proposed methods.
2. Missing the discussion of the related works and baselines which also focus on text to molecule generation based on large language models [1]. [2]; Missing the discussion of the related works about text-to-graph diffusion models [3], [4].
3. The experimental results are not promising. In text-guided molecule generation, the proposed method only outperforms the smallest model, MolT5-small, on three metrics. Although the model has a small parameter size, its utility performance is significantly worse than that of current models in this domain, which limits its practical applications.

[1] Unifying Molecular and Textual Representations via Multi-task Language Modelling.

[2] Mol-Instructions: A Large-Scale Biomolecular Instruction Dataset for Large Language Models.

[3] Text-Guided Molecule Generation with Diffusion Language Model

[4] 3M-Diffusion: Latent Multi-Modal Diffusion for Language-Guided Molecular Structure Generation

**Questions:**

1. Can the authors talk more about their novelty compared with existing text-to-graph diffusion models as mentioned in the weakness part?

---

> ### Author Response · Authors · 2024-11-25
>
> Thank you for your detailed review and thoughtful comments.
>
> Please find our responses to your points below:
>
> **Q:** More datasets.
>
> **A:** ChEBI-20 is currently the most representative dataset for the text-to-molecule generation task, containing 33,010 molecule-description pairs. As demonstrated in works [1,3,5,6], this dataset is the most commonly used in related research. In addition to molecule generation, we have also developed a text-to-graph dataset that focuses exclusively on topological structures and conducted comprehensive experiments on it. Our method achieves superior generation performance compared to existing LLMs.
>
> **Q:** The discussion of the related works about text to molecule generation method and text-to-graph diffusion models.
>
> **A:**
> [1] and [2] are text-to-molecule methods that leverage language models for generation, where molecules are represented in the SMILES format. [3] employs a text diffusion model to generate the SMILES representation of molecules. While these methods focus on generating molecules from text representations, they do not extend to other graph generation tasks. [4] utilizes a latent space diffusion model to generate latent features, which are then decoded into molecule graphs using a graph decoder. In contrast, our work is the first to explore a general approach for text-to-graph generation. We have included these discussions and references in the related work section of the revised version of the paper.
>
> [1] Unifying Molecular and Textual Representations via Multi-task Language Modelling. ICML 2023.
>
> [2] Mol-Instructions: A Large-Scale Biomolecular Instruction Dataset for Large Language Models. ICLR 2024.
>
> [3] Text-Guided Molecule Generation with Diffusion Language Model. AAAI 2024.
>
> [4] 3M-Diffusion: Latent Multi-Modal Diffusion for Language-Guided Molecular Structure Generation. COLM 2024.
>
> [5] MolT5: Translation between Molecules and Natural Language. EMNLP 2022.
>
> [6] BioT5: Enriching Cross-modal Integration in Biology with Chemical Knowledge and Natural Language Associations. EMNLP 2023.
>
> **Q:** The explanation of the experimental results.
>
> **A:** The experimental results for MolT5 presented in this paper are based on MolT5-base, which has 220M parameters, rather than MolT5-small. As mentioned earlier, we use a graph diffusion model for molecule generation. Due to limitations in model architecture and data scale, there is an inherent semantic gap between graph-based models and text-based models.
> Nevertheless, leveraging the graph-aligned LLM and the structure-aware cross-attention mechanism we propose, our text-to-graph generation method achieves comparable performance to mainstream text-to-molecule methods in molecular semantic metrics on the molecule generation task. Furthermore, thanks to the inherent characteristics of the graph diffusion model, our method demonstrates superior validity and diversity in the generated results compared to the baseline language models, despite using a significantly smaller model size.
>
> **Q:** Our novelty compared with existing text-to-graph diffusion models.
>
> **A:** As noted earlier, existing works primarily focus on text-to-molecule tasks rather than the broader text-to-graph domain. Our method is the first to utilize a graph diffusion model for generating graph-structured data in a text-to-graph setting, introducing a general framework for text-to-graph generation. This framework integrates large language models (LLMs) with graph diffusion models.
> We propose a novel approach that includes self-supervised text-graph alignment to improve LLMs' comprehension of graph structures and a structure-aware cross-attention mechanism to guide diffusion models in generating diverse and relationally accurate graphs.

---

> > ### Comment · Reviewer_JV2T · 2024-11-28
> >
> > Thanks for authors' rebuttal! However, I find the level of novelty insufficient compared to previous methods. For instance, while the authors claim that their approach is the first to use a graph diffusion model, [1] also employs latent graph diffusion models, which limits the uniqueness of the contribution. Additionally, the alignment technique has already been utilized in works such as [1] and [2].
> >
> > [1] 3M-Diffusion: Latent Multi-Modal Diffusion for Language-Guided Molecular Structure Generation. COLM 2024.
> >
> > [2] Text-Guided Molecule Generation with Diffusion Language Model. AAAI 2024.

---

### Official Review · Reviewer_rbdC · 2024-11-04

**Soundness:** 2
**Presentation:** 3
**Contribution:** 3
**Rating:** 5
**Confidence:** 5

**Summary:**

The paper proposes a novel approach that combines large language models (LLMs) with graph diffusion models for graph construction based on input text. Specifically, the method first pretrains an LLM to learn graph generative features in a self-supervised manner, aimed at predicting structural characteristics of a graph. Once the LLM is pretrained, its parameters are frozen, and it is utilized to generate representations of the conditioning text. A structure-aware cross-attention mechanism then allows a graph transformer to approximate the score function of a graph diffusion model grounded in stochastic differential equations.

**Strengths:**

- The authors address an interesting problem of text-to-graph construction using an LLM to guide a graph diffusion model.
- Since the LLM in this method is fixed after pretraining, it doesn’t impose much complexity overhead that makes training the diffusion model more feasible.
- Authors utilize informative visualizations which help to understand the paper.
- The method shows superior performance in most of the graph generation experiments.

**Weaknesses:**

- The objectives used for pretraining the LLM may be too simplistic, as tasks such as predicting the number of nodes, edges, or subgraphs could be reduced to mere counting problems for the model. This focus on counting may not effectively encourage the LLM to learn meaningful semantics, potentially limiting its generalization and robustness. Given that the LLM is responsible for generating informative representations of tokens, the authors could consider incorporating auxiliary objectives that promote understanding of token semantics and their dependencies, such as contrastive learning between connected and unconnected nodes.
- In the literature, attention scores typically refer to matrices where the row-wise sums equal 1, making the designation of the matrix $A_{edge, uv}$ as an attention matrix potentially misleading. The authors should either clarify this point or normalize the matrix so that the row sums equal one. Alternatively, using a different term for the matrix could also be a viable option.
- While the graph diffusion process is introduced at the beginning of the "Preliminaries," the subsequent processing steps following the generation of the matrices $X_{cond}$ and $E_{cond, uv}$ in the diffusion workflow are not discussed. Providing a detailed explanation of the steps that follow the construction of these matrices would enhance clarity in the framework and improve the reproducibility of the method.
- The authors have not provided their code or sufficient implementation details for the method, raising concerns about the reproducibility of the results.
- The authors rely on a single molecular dataset as their only real-world dataset for experiments. Including additional experiments on real-world datasets from other domains would provide a better evaluation of the method's generalization, especially since it has been outperformed on this dataset based on three out of six metrics.

**Questions:**

- Regarding Equation 8, the dimensionality of matrix $C$ of token representations must match with nodes feature matrix $X$. However, how the dimensionality of matrix $C$ is set for compatibility with $X$ is not discussed. In other words, there must be a process of detecting tokens related to nodes in the input text so that their embeddings represent the node. Is there an entity recognition step employed before extracting the textual representation?
- Regarding Equation 12, the edge conditioning matrix $E_{cond, uv}$ is composed of continuous elements meaning that it represents a complete graph. However, usually, molecular graphs are sparse, and making complete graphs increases complexity and would also lead to over-smoothing issues. How do authors encourage the sparsity of the graphs?

---

> ### Author Response · Authors · 2024-11-25
>
> Thank you for your detailed review and valuable comments.
>
> Please find our responses to your points below:
>
> **Q:** The objectives used for pretraining the LLM.
>
> **A:** As highlighted in [1,2], self-supervised tasks are known to enhance the performance of neural networks. In our work, tasks such as predicting the number of nodes, edges, or subgraphs help the LLM develop a graph-centric understanding of the input text, enabling the extracted features to effectively guide the graph generation process. The ablation study in Table 4 further confirms that our fine-tuned LLM outperforms the original LLM in guiding the graph diffusion model for generation. We appreciate your thoughtful suggestion regarding alternative pretraining objectives and will consider it for future exploration.
>
> [1] Graph Self-Supervised Learning: A Survey. TKDE 2022.
>
> [2] Self-Supervised Learning of Graph Neural Networks: A Unified Review. TPAMI 2022.
>
> **Q:** Explanation of Equation 10 regarding the question about the matrix $A_{edge,uv}$.
>
> **A:** Thank you for your careful review. We noticed a typo in Eq. (10) and have corrected it. The correct equation is $G_{2,uv} = 1 - G_{1,uv}$, which ensures that the result in Eq. (11) is properly normalized over the text tokens. We have updated this in the revised version of the paper. We appreciate your attention to detail.
>
> **Q:** Providing a detailed explanation of the steps that follow the construction of these matrices
>
> **A:** Our conditional graph diffusion model works on a tensor representation of graphs, where the node types $X$ and adjacency matrix $E$ are encoded into $X_{enc} \in \mathbb{R}^{N \times C_{node}}$ and $E_{enc} \in \mathbb{R}^{N \times N \times (C_{edge} + 1)}$ respectively using one-hot encoding.
> $$X_{enc} = one\_hot(X), E_{enc} = one\_hot(E)$$
> To generate new graphs, we sample $X_{enc}$ and $E_{enc}$ using the trained graph diffusion model, and convert them into graphs by taking the argmax of $X$ and $E$ as the node types and adjacency matrix.
> $$X = argmax(X_{enc}), E = argmax(E_{enc})$$
> As for $X_{cond}$ and $E_{cond,uv}$, they indicate the outputs of the structure-aware cross-attention mechanism, and they will be used in the score predictor of the graph diffusion model.
>
> **Q:** Code or sufficient implementation details.
>
> **A:**
> For the graph-aligned LLM, we use Llama-3-8B and fine-tune it on the graph structure prediction task using LoRA (rank=8, alpha=64) over 6 epochs.
> For the conditional graph diffusion model, we adopt the GDSS-Transformer implementation as the base architecture for the score predictor and integrate a structure-aware cross-attention module at every layer of the graph transformer. This modification enables text-conditional graph generation. The number of layers and feature channels remain consistent with those used in GDSS-Transformer.
> For molecule generation, the model is initialized with a pre-trained checkpoint on the ZINC250k dataset and subsequently fine-tuned with a learning rate of 2e-4 on the ChEBI-20 dataset for 1000 epochs. For the synthetic dataset, the model is trained with a learning rate of 1e-4 for 1000 epochs.
> Further details on the methodology and experimental settings, along with the associated codes and models, will be made publicly available on GitHub upon acceptance of the paper.
>
> **Q:** More datasets.
>
> **A:** ChEBI-20 is currently the most recognized dataset for the text-to-molecule generation task, containing 33,010 molecule-description pairs. As demonstrated in works [1,2,3,4], this is the only dataset used in similar studies. Beyond molecule generation, we also constructed a text-to-graph dataset focused solely on topological structures and conducted extensive experiments on it. Compared to current LLMs, our method achieves better generation performance.
>
> [1] Unifying Molecular and Textual Representations via Multi-task Language Modelling. ICML 2023.
>
> [2] Text-Guided Molecule Generation with Diffusion Language Model. AAAI 2024.
>
> [3] MolT5: Translation between Molecules and Natural Language. EMNLP 2022.
>
> [4] BioT5: Enriching Cross-modal Integration in Biology with Chemical Knowledge and Natural Language Associations. EMNLP 2023.
>
> **Q:** The explanation of Equation 8.
>
> **A:** The shape of $C$ is [number of tokens, text feature dim], and the shape of $X$ is [number of nodes, node feature dim]. We can select $W$ with appropriate dimensions so that $Q$ and $K$ have the same size in the second dimension, avoiding any dimension mismatch issues.
>
>
> **Q:** The explanation of Equation 12.
>
> **A:** The edge conditioning matrix is computed based on graphs generated during the diffusion process, which have Gaussian noise injected into their adjacency matrices, resulting in inherently dense graphs. This is a characteristic of score-based graph diffusion. Exploring alternative diffusion frameworks that can preserve graph sparsity is an interesting direction that we leave for future work.

---

> ### Comment · Reviewer_rbdC · 2024-11-28
> **Response to author's rebuttal**
>
> Thank you authors for your rebuttal.
>
> However, my concerns about the reproducibility of results and generalization to more real-world datasets remain. Therefore, I'd keep my score.

---

### Meta-Review · Area_Chair_h6pY · 2024-12-16

**Metareview:**

The paper investigates an interesting problem of text-to-graph construction using an LLM, which is well motivated. The paper is well written and organized to aid understanding.

However, there are some concern on the reproducibility of experiments and the generalization to more real-world datasets. The novelty of the proposed approach is not well articulated, especially the differentiation from existing graph diffusion and alignment models.

**Additional Comments On Reviewer Discussion:**

While the authors tried to respond to the feedback, reviewers still find the following points not well addressed, which I am agreeable.

1. Code is not available for reproducibility check. (Authors only promised to release after acceptance)
2. There is limited evidence of generalizing to other real-world datasets.
3. Novelty of the proposed approach is not clearly explained, esp. w.r.t. previous diffusion and alignment models.

---

### Decision · Program_Chairs · 2025-01-22

Reject